# Foot-and-Mouth Disease Virus Evades Innate Immune Response by 3C-Targeting of MDA5

**DOI:** 10.3390/cells10020271

**Published:** 2021-01-29

**Authors:** Hyejin Kim, Ah-Young Kim, Jieun Choi, Sun Young Park, Sang Hyun Park, Jae-Seok Kim, Sim-In Lee, Jong-Hyeon Park, Choi-Kyu Park, Young-Joon Ko

**Affiliations:** 1Animal and Plant Quarantine Agency, Gimcheon-si 39660, Korea; kiss86j@naver.com (H.K.); mochsha@korea.kr (A.-Y.K.); dr98cju@naver.com (J.C.); sun3730@korea.kr (S.Y.P.); shpark0205@korea.kr (S.H.P.); kimjs0728@korea.kr (J.-S.K.); lunark2@korea.kr (S.-I.L.); parkjhvet@korea.kr (J.-H.P.); 2College of Veterinary Medicine, Animal Disease Intervention Center, Kyungpook National University, Daegu 41566, Korea

**Keywords:** foot-and-mouth disease virus, MDA5, non-structural protein, 3C, innate immune response

## Abstract

Foot-and-mouth disease (FMD) is a highly contagious disease caused by FMD virus (FMDV) in cloven-hoofed animals. Retinoic acid-inducible gene I (RIG-I) and melanoma differentiation-associated gene 5 (MDA5) are representative receptors in the cytoplasm for the detection of viral RNA and trigger antiviral responses, leading to the production of type I interferon. Although MDA5 is a crucial receptor for sensing picornavirus RNA, the interplay between MDA5 and FMDV is relatively unknown compared to the interplay between RIG-I and FMDV. Here, we observed that the FMDV infection inhibits MDA5 protein expression. Of the non-structural proteins, the Lb and 3C proteinases (Lb^pro^ and 3C^pro^) were identified to be primarily responsible for this inhibition. However, the inhibition by 3C^pro^ was independent of proteasome, lysosome and caspase-dependent pathway and was by 3C protease activity. A direct interaction between 3C^pro^ and MDA5 protein was observed. In conclusion, this is the first report that 3C^pro^ inhibits MDA5 protein expression as a mechanism to evade the innate immune response during FMDV infection. These results elucidate the pathogenesis of FMDV and provide fundamental insights for the development of a novel vaccine or therapeutic agent.

## 1. Introduction

The Foot-and-Mouth Disease virus (FMDV) is a single-stranded, positive-sense RNA virus that belongs to the genus *Apthovirus* of the family *Picornaviridae* [1]. The FMDV causes foot-and-mouth disease (FMD), which is highly contagious in various cloven-hoofed animals, such as pigs, cows and goats [2,3]. The FMDV genome is 8.5 kb and consists of one open reading frame (ORF), which encodes a single polyprotein that is subsequently proteolyzed into four structural proteins (SPs: VP1, VP2, VP3 and VP4) and eight nonstructural proteins (NSPs: L^pro^, 2A, 2B, 2C, 3A, 3B, 3C^pro^ and 3D) [2,4,5].

The innate immune system plays an important role in defending against invading pathogens. Pathogen recognition receptors (PRRs) recognize viral pathogen-associated molecular patterns (PAMPs) in host cells and activate signaling cascades, leading to the expression of the host immune response, type 1 interferons (IFNs; alpha/beta interferon [IFN-α/β]) genes and pro-inflammatory cytokines [6]. PRRs involved in RNA viral genome detection include Toll-like receptor (TLR) and retinoic acid-inducible gene I (RIG-I)-like receptor (RLR) [7]. TLRs are expressed on the surface or endosomal compartments of macrophages, dendritic cells and other immune cells [8]. The RLR family mainly encompasses two members: RIG-I and melanoma differentiation-associated gene 5 (MDA5), which are RNA sensors that play an important role in inducing the immune defense against RNA virus infection [9,10].

RIG-I and MDA5 have two amino-terminal caspase activation and recruitment domains (CARDs) at the N-terminus, a central DExD/H box ATPase/helicase domain and a regulatory/repression domain at the C-terminus [11]. Following activation after viral RNA recognition, the CARDs of RIG-I or MDA5 undergo conformational changes to be exposed and multimerized, which further allows CARD–CARD interactions with mitochondrial antiviral-signaling protein (MAVS). The MAVS then relays the signal to TANK-binding kinase 1 (TBK1) and IκB kinase-ε (IKKε), which subsequently leads to the activation of interferon regulatory factor 3 (IRF3) and nuclear factor-κB (NF-κB) transcription factor [12]. Their activity ultimately induces the expression of type I-IFN and pro-inflammatory cytokine production, leading to the host antiviral signaling cascades [13]. Although RIG-I and MDA5 are similar proteins that induce type I IFN production, they appear to specialize in the detection of different viruses [14]. While RIG-I is essential for detecting infection of many negative-strand RNA viruses and some flaviviruses [15,16,17,18], MDA5 is critical for the recognition of the picornavirus, coronavirus and calicivirus families [19,20,21]. However, many studies have focused on RIG-I compared to MDA5 in relation to the FMDV of the family picornavirus [22,23,24,25], little is known about the interplay between MDA5 and FMDV.

The aim of this study was to investigate mechanism by which FMDV evades the host innate immune response. Thereby, we introduce a new tactic of FMDV to evade the host innate immune response, which might shed new light on antiviral research.

## 2. Materials and Methods

### 2.1. Cells and Viruses

Porcine Kidney-15 (PK-15; ATCC CCL-33, Manassas, VA, USA) cells, human embryonic kidney 239T (HEK 293T; ATCC, Manassas, VA, USA) cells and baby hamster kidney-21 (BHK-21; ATCC C-13, Manassas, VA, USA) cells were grown in Dulbecco Modified Eagle’s medium (DMEM; Corning Inc., Corning, NY, USA) containing 10% fetal bovine serum (FBS; Gibco, Grand Island, NY, USA) and 1% antibiotic-antimycotic (Anti-Anti; Gibco) at 37°C with 5% CO_2_. The BHK-21 cell is commonly used for FMDV propagation and titration. The FMDV serotype O strain (Boeun/SKR/2017) was propagated in BHK-21 cells. The FMDV type O strain was inoculated at a multiplicity of infection (MOI) of 0.5 on PK-15 cells of 70–80% confluency and was incubated at 37 °C with 5% CO_2_. Viral titers were calculated by endpoint titration on BHK-21 cells using the Spearman-Kärber method and the results were expressed as log_10_ TCID_50_/0.1 mL [26,27].

### 2.2. Reagents

The commercial antibodies for target proteins used in this study were as follows: anti-HA-tag polyclonal antibody (MBL, Nagoya, Japan), anti-HA-tag monoclonal antibody (Cell Signaling Technology, Danvers, MA, USA), anti-MDA5 polyclonal antibody (Abcam, Cambridge, UK), anti-MAVS polyclonal antibody (Abcam), anti-IRF3 polyclonal antibody (Cell Signaling Technology), phospho-IRF3 monoclonal antibody (Cell Signaling Technology), OctA (FLAG)-Probe monoclonal antibody (Santa Cruz Technology, Dallas, TX, USA), anti-mouse IgG (Cell Signaling Technology), anti-rabbit IgG (H+L) secondary antibody (Invitrogen, Meridian Rd., Rockford, IL, USA), anti-mouse IgG (H+L) secondary antibody (Invitrogen), anti-mouse IgG-Fc Fragment antibody (Bethyl Laboratories, Montgomery, Texas, USA) and anti-β-actin polyclonal antibody (Cell Signaling Technology). The chemical reagents used for the inhibitor assay used in this study include: the proteasome inhibitor MG132 (Millipore, Billerica, MA, USA), the lysosome inhibitor, chloroquine diphosphate (CQ; Sigma, St. Louis, MO, USA) and the caspase inhibitor, benzyloxy-carbonyl-Val-Ala-Asp (OMe) fluoromethylketone (Z-VAD-FMK; Merck, Darmstadt, Germany).

### 2.3. Plasmids

To generate expression vectors for FMDV Lb (GenBank accession No. AJ539139), 2B (GenBank accession No. AY312587) and 2C (GenBank accession No. AY312587), 3A (GenBank accession No. AY312587) and 3B (GenBank accession No. AY312587) and 3C (GenBank accession No. AJ539139) or 3D (GenBank accession No. AJ539139), each NSP cDNA was cloned into the pCMV-HA vector to construct plasmids expressing HA-tagged protein (Enzynomics, Daejeon, Korea). Similarly, the cDNA of porcine MDA5 (GenBank accession No. EU006039) was cloned into the pCMV-HA vector or p3xFLAG-CMV vector (Enzynomics). Specific mutations of 3C (H46Y, D84N and C163G) were introduced to pCMV-HA-3C (Enzynomics).

### 2.4. Transfection

HEK293T cells, approximately 70–80% confluent in 6-well plates on the day of transfection, were washed once with pre-warmed phosphate-buffered saline (PBS; Corning Inc.) and were completely replaced by Opti-MEM I (1×) Reduced Serum Medium (Invitrogen, Grand Island, NY, USA) per plate, followed by 30 min incubation at 37 °C. According to the manufacturer’s instructions for FuGENE HD Transfection Reagent (Promega, Madison, WI, USA), the transfection reagent and plasmid DNA were mixed gently at a ratio of 3:1. The transfection reagent/DNA mixture was incubated at room temperature (RT) for 15 min and added to the plates. To assay proteasome, lysosome and caspase inhibition, media of transfected cells were supplemented with the proteasome inhibitor MG132, the lysosome inhibitor CQ and the caspase inhibitor Z-VAD-FMK, respectively. After 4 h post-transfection (hpt), the medium was replaced with 2% FBS and the plates were incubated in an incubator at 37 °C with 5% CO_2_.

### 2.5. RNA Extraction and Real-Time RT-PCR

Total RNA was extracted from FMDV-infected PK-15 cells or plasmid-transfected HEK293 cells using the RNeasy mini kit (Qiagen, Valencia, CA, USA) and the cDNA was synthesized by the Super Script III First-Strand Synthesis System (Invitrogen, Carlsbad, CA, USA) from the extracted RNA, according to the manufacturer’s protocol. The level of mRNA for FMDV was detected using an AccuPower FMDV Real-time RT-PCR Kit (Bioneer, Daejeon, Korea). To quantify the abundance of mRNA, the quantitative real-time RT-PCR experiment was performed with iQ SYBR Green Super mix (Bio-Rad, Hercules, CA, USA) on a CFX96 Real-Time System (Bio-Rad, Munich, Germany) under the following conditions: 95 °C for 5 min, followed by 40 cycles of 95 °C for 10 s and 60 °C for 30 s. The Glyceraldehyde-3-Phosphate dehydrogenase (GAPDH) gene was used as an internal control. Relative expression of mRNA expression was determined by the threshold cycle (2^−ΔΔCT^) method. The primer sequences for target gene expression are listed in Table 1 [28]. 

### 2.6. Immunoprecipitation (IP) and Western Blot Analysis

The FMDV-infected PK-15 and plasmid-transfected HEK293T cells were washed three times with pre-warmed PBS (Corning Inc.) and lysed in chilled Pierce radioimmunoprecipitation assay buffer (RIPA buffer; Thermo Fisher Scientific, Rockford, IL, USA) or Pierce IP lysis buffer (Thermo Fisher Scientific) containing protease inhibitor cocktail (Complete Mini, Roche Diagnostics, Germany) and phosphatase inhibitor (Thermo Fisher Scientific) using a cell scraper on ice. To remove the cell debris, the lysates were centrifuged at 13,000× *g* for 20 min at 4 °C and the supernatant was transferred to a new tube for further analysis. Total protein concentrations were determined using the Pierce™ BCA Protein Assay Kit (Thermo Fisher Scientific, Waltham, MA, USA) according to the manufacturer’s instructions.

For immunoprecipitation (IP), 500 µL of whole cell lysate was incubated with 2 µg of OctA (FLAG)-Probe monoclonal antibody or mouse IgG antibody overnight at 4 °C with shaking. The next day, the lysates were incubated with 100 µL of protein A/G agarose resin (Thermo Fisher Scientific) at 4 °C for 2 h and centrifuged 3000× *g* for 1 min to pellet resin. The agarose resin was washed five times with 500 µL of IP lysis buffer containing protease inhibitor cocktail and the antigen-antibody complex is eluted from the resin by heating in 100 µL of SDS loading buffer without Sample Reducing agent (Invitrogen) for 10 min at 50 °C and centrifuged 3000× *g* for 1 min.

For Western blot analysis, the samples were mixed with 4× lithium dodecyl sulfate sample buffer (LDS; Invitrogen) containing Sample Reducing agent (Invitrogen) and were then heated for 10 min at 95 °C. Target proteins were separated on 4–12% Bis-Tris gels (Invitrogen) and transferred onto a PVDF membrane (Invitrogen) using the iBlot gel transfer device (Invitrogen). The membranes were blocked with 2% skim milk in PBS containing 0.1% Tween 20 (PBS-T) for an hour at RT with shaking, washed three times with PBS-T for 10 min and then incubated with appropriate primary antibodies overnight at 4 °C with shaking. The next day, the membranes were washed three times with PBS-T and incubated with suitable secondary antibodies for an hour at RT with shaking. The antibody-antigen complexes were visualized with ECL Western blotting substrate (Amersham, Buckinghamshire, UK) using the Azure C600 device (Azure Biosystem, Dublin, CA, USA). Band intensities were quantified using Image J software (Wayne Rasband, NIH, Bethesda, MD, USA).

### 2.7. Statistical Analysis

The statistical significance was evaluated with one-way ANOVA followed by Tukey’s *post hoc* test using GraphPad Prism Version 5 (GraphPad Software, San Diego, AD, USA). All data are representative of three independent experiments and values are represented as the mean ± standard error of the mean (SEM). *P*-values < 0.05 were considered statistically significant.

## 3. Results

### 3.1. FMDV Infection Induces MDA5, RIG-I and IFN Transcription but Inhibits MDA5 Protein Expression in PK-15 Cells

To determine whether FMDV infection induced of IFN-β production through RLR-mediated antiviral pathways, PK-15 cells were infected with FMDV at a multiplicity of infection (MOI) of 0.5 for 0, 4, 8, 12, 16 and 24 h (Figure 1). The mRNA and protein levels were examined by real-time RT-PCR and western blot analysis. FMDV mRNA levels and viral titers were highest at 12 h post-infection (hpi) and decreased from 16 hpi (Figure 1a). The mRNA levels of MDA5, RIG-I and IFN-β gradually increased and were highest at 16 hpi (Figure 1b). The levels of the three mRNAs decreased after 16 hpi. The protein expression levels of MDA5, RIG-I and MAVS gradually decreased in a time-dependent manner as FMDV replicated in the PK-15 cells. In particular, the expression of the MDA5 protein was significantly reduced 16 hpi compared to that of the RIG-I protein (Figure 1c). While the amount of IRF3 was not affected by FMDV infection in PK-15 cells, the phosphorylated form of IRF3 (pIRF3), which induces IFN-β transcription, increased until 16 hpi and decreased at 24 hpi (Figure 1d). These results suggest that MDA5 plays a more important role than RIG-I in innate immune evasion of FMDV.

### 3.2. Lb^pro^ and 3C^pro^ of FMDV NSPs Reduce Endogenous and Exogenous MDA5 Expression

FMDV infection contributed to the decline in MDA5 protein expression. Therefore, to identify the FMDV NSP that affect the expression of endogenous MDA5 and RIG-I proteins, HEK293T cells were transfected with the plasmid encoding each viral NSP. After 16 h, endogenous MDA5 and RIG-I were detected by western blotting in the transfected cells (Figure 2a) and quantified using ImageJ software (Figure 2b). The Lb^pro^ and 3C^pro^ of FMDV NSPs significantly reduced endogenous MDA5 expression compared to HA-vector-transfected cells. Meanwhile, RIG-I expression was not affected by any type of FMDV NSP. To identify whether the Lb^pro^ and 3C^pro^were responsible for the decline in exogenous MDA5 protein, plasmids encoding HA-tagged MDA5 and NSPs were co-transfected into HEK293T cells for 16 h (Figure 3a). MDA5 was not detected in the cells transfected with Lb^pro^ and 3C^pro^ by western blot analysis. MDA5 protein expression was reduced in a dose-dependent manner by Lb^pro^ and 3C^pro^ (Figure 3b). MDA5 expression was also reduced by 2B, 2C and 3A but not by 3B and 3D. The inhibition by 2B, 2C and 3A was dose-dependent for exogenous MDA5 (Figure 3c). In particular, the Lb^pro^ and 3C^pro^ treatments decreased MDA5 mRNA levels (Figure 3b). These results indicate that both endogenous and exogenous MDA5 protein levels were prominently reduced by Lb^pro^ and 3C^pro^.

### 3.3. FMDV Lb^pro^-or 3C^pro^-Induced Reduction of MDA5 is Independent of Proteasome, Lysosome and Caspase Pathways

To determine whether the proteasome, lysosome and caspase-dependent pathways play roles in the FMDV Lb^pro^ and 3C^pro^-induced reduction of MDA5, the effects of proteasome inhibitor (MG132), lysosome inhibitor (CQ) and caspase inhibitor (Z-VAD-FMK) were evaluated (Figure 4). MDA5-expressing plasmids and either HA-Lb or HA-3C plasmids were co-transfected into HEK293 cells and the cells were cultured in the presence or absence of the inhibitors. The protein levels of MDA5, Lb^pro^ and 3C^pro^ were detected by western blotting at 12 hpt. MG132, CQ or Z-VAD-FMK did not affect the L-or 3C-induced reduction of MDA5 (Figure 4). These results suggest that FMDV Lb^pro^ and 3C^pro^-induced reduction of MDA5 occurs via a process independent of the proteasome, lysosome and caspase pathways.

### 3.4. The Catalytic Residues in 3C^pro^ Active Sites Are Essential for 3C^pro^-Induced MDA5 Reduction

The catalytic triad of H46, D84 and C163 in FMDV 3C^pro^ have been determined as crucial sites that play essential roles in its enzyme activity [29,30]. To determine whether the enzyme activity of 3C^pro^ was involved in MDA5 reduction, we constructed three mutant plasmids that expressed the 3C^pro^ mutant (HA-3C-H46Y, HA-3C-D84N and HA-3C-C163G), which eliminated the protease activity of 3C^pro^ (Figure 5). The pCMV-HA-vector plasmids, HA-3C plasmids or mutant plasmids were co-transfected with Flag-MDA5 plasmids into HEK293T cells and the expression levels of Flag-MDA5 were detected at 12 hpt (Figure 5a). The wild-type HA-3C^pro^ (WT) strongly suppressed MDA5 protein expression, compared to the pCMV-HA-Vector (Figure 5b). In contrast, the H46Y, D84N and C163G mutants had no suppressive effect on MDA5 expression, demonstrating that the protease activity of 3C^pro^ inhibited MDA5 protein expression (Figure 5b). An immunoprecipitation was then performed to identify whether the interaction between 3C^pro^ and MDA5 was associated with the 3C protease activity of (Figure 6). The wild-type and the mutants of HA-3C^pro^ interacted with FLAG-MDA5 and the wild-type of 3C^pro^ inhibited MDA5 protein expression. Meanwhile, the D84N and C163G mutants did not reduce MDA5 protein expression. These results suggest that 3C^pro^ directly interacts with MDA5 and induces the reduction of MDA5 by its protease activity.

## 4. Discussion

Upon infection of the host, the virus will face an attack from the host immune response. In the battle with the host immune response, the virus has evolved a series of immune escape mechanisms to overcome the antiviral responses induced by the host immune system [31]. Although MDA5 is known to be responsible for picornavirus recognition [32,33], little is known about the interplay between MDA5 and FMDV. This study aimed to elucidate the mechanism of FMDV in evading the immune response of the host and to define specific NSPs that play a key role in the interplay with MDA5. 

The evidence that FMDV interferes with interferon signaling cascade pathways is supported by two results in this study. Firstly, FMDV infection degraded MDA5 and MAVS. Secondly, IRF3 phosphorylation, which is a hallmark of IRF3 activation, increased until 16 h post-FMDV infection and then decreased that is similar pattern to the mRNAs for interferon beta, MDA5 and RIG-I. In this study, the PK-15 cells were employed to see phenomena related with interferon signal transduction after FMDV infection because the PK-15 was susceptible to FMDV and have interferon signaling pathway [34,35,36]. Endogenous MDA5 in PK-15 decreased significantly on FMDV infection, resulting in no protein band by western blot analysis (Figure 1). Previous studies have shown that MDA5 is inhibited by 3A or 2B. They, however, were carried out using Sendai virus [11] or a plasmid expressing MDA5 [37]. This study is the first to report that endogenous MDA5 expression is inhibited by FMDV infection. The result that MDA5 mRNA increased and MDA5 protein decreased until 16 hpi indicates that protein level rather than transcription was interfered with by FMDV infection. Once FMDV enter the host’s cell, interferon signaling pathway initiates after MDA5 recognizes FMDV RNA, leading to increasing MDA5 mRNA. Meantime, NSP that is accumulated as FMDV infection progress in the cells interferes with interferon signaling pathway by degrading MDA5 protein translated from mRNA. We suggest that that is why the amount of MDA5 between mRNA and protein level was different. In previous studies, the similar result was reported by other research group that FMDV infection can induce RIG-I mRNA expression, while RIG-I protein was gradually downregulated as an infection progress [24].

This study showed that MDA5 was more significantly reduced than RIG-I by FMDV infection in PK-15 cells. Previous study reported that FMDV was related with MDA5 but not by TLR3 in porcine epithelial cells using lentivirus-driven RNA interference. Although a previous study showed that MDA5 overexpression showed more significant antiviral activity than RIG-I overexpression against FMDV [24], it dealt with the interplay between 2B and RIG-I only. Another study reported that L^pro^ cleaved exogenous MDA5 but it did not show any evidence of degradation of endogenous MDA5 [38].

Since it was found that FMDV evaded interferon induction pathways mainly via MDA5 inhibition in this study, we investigated which NSP could play a key role in the evasion mechanism. A total of seven NSPs (excluding 2A, which directs co-translational “ribosome skipping” at its own C-terminus [39]), were evaluated for the interplay with MDA5 by overexpressing each protein. Of these, L^pro^ and 3C^pro^ significantly inhibited endogenous MDA5 expression in contrast to RIG-I, which was not reduced by any of them. The slight decrease in RIG-I caused by FMDV infection (Figure 1c) might be attributed to the structural proteins such as VP1 or VP3 that were reported previously with regard to the interferon signaling pathway [28,40].

In addition, exogenous MDA5 expression was evaluated with the same NSPs using HA-tagged plasmids. Consistent with endogenous MDA5, L^pro^ and 3C^pro^ inhibited the most significant exogenous MDA5 expression. Other NSPs, such as 2B, 2C and 3A, also inhibited exogenous MDA5 compared to 3B and 3D. The contradictory results of 2B, 2C and 3A for endogenous and exogenous MDA5 might be due to the fact that endogenous MDA5 was derived from humans (HEK293T) and the exogenous MDA5 plasmid was constructed based on the porcine sequence. In this study, HEK293T cells were used due to its much higher transfection efficiency than that of PK-15 cells. Although this study focused on L^pro^ and 3C^pro^, the interplay of 2B, 2C and 3A with MDA5 still has to be investigated.

In response to FMDV infection, the proteasome, lysosome and caspase-dependent pathways were not involved in the reduction of MDA5 by 3C^pro^. Similar results were previously reported with RIG-I, which was degraded during FMDV, EMCV, poliovirus and rhinovirus infection [24,41]. Since we assumed that the protease activity of 3C would cleave MDA5, 3C mutants with point mutations in the catalytic triad of H46, D84 and C163 were constructed. As anticipated, all three inactive 3C^pro^ mutants (H46Y, D84N and C163G) did not cleave the MDA5 at all. Therefore, we performed immunoprecipitation assay using HA-3C and Flag-MDA5 to observe the direct interaction between 3C^pro^ and MDA5. The 3C^pro^ interacted with MDA5 and that its protease activity degraded MDA5 protein expression. In this study, the degradation product of 3C^pro^-induced MDA5 was not detected. Previous studies reported that Lb^pro^ degraded MDA5 without degradation products [35] and that 2B degraded RIG-I without intermediate products [24]. In further studies, we are planning to research regarding its degradation mechanism because the MDA5 was degraded independent of the proteasome, lysosome and caspase pathways. FMDV L^pro^ is expressed in two forms, Lab and Lb. Lb is more abundant than Lab in infected cells [42] and is associated with translocation to the nucleus and cleavage of NF-kB and IRF3/7 proteins [43,44,45]. Since the direct interaction between the FMDV Lb^pro^ and MDA5 has recently been reported [38], we focused on the interplay between FMDV 3C^pro^ and MDA5, which has not been elucidated at all.

FMDV 3C^pro^ specializes in cleaving FMDV polyprotein into viral proteins with biological functions and induces cleavage of eIF4A and histone H3 to block the host translation system [46,47]. To antagonize the interferon signaling pathway, the protease activity of FMDV 3C^pro^ blocks STAT1/STAT2 nuclear translocation [36]. These biological functions of FMDV 3C^pro^ appear to ruin the IFN system in the broad spectrum [48]. However, there is no report that FMDV 3C^pro^ could exert a role to inhibit innate immunity by targeting MDA5.

## 5. Conclusions

FMDV infection initially triggers interferon production via IRF3 phosphorylation but eventually inhibits the expression of the MDA5 protein, thereby disrupting p-IRF3 activity. In this regard, we identified the antagonistic role of FMDV Lb^pro^ and 3C^pro^ on MDA5-mediated immune responses. In particular, the protease activity of 3C^pro^ inhibited MDA5 expression independent of other protein degradation pathways. To our knowledge, this is the first report that FMDV 3C^pro^ targets MDA5 in the mechanism to evade the immune response of the host. These results will improve our understanding of FMDV pathogenesis in the host and provide fundamental insights for the development of novel vaccines or therapeutic agents.

## Figures and Tables

**Figure 1 cells-10-00271-f001:**
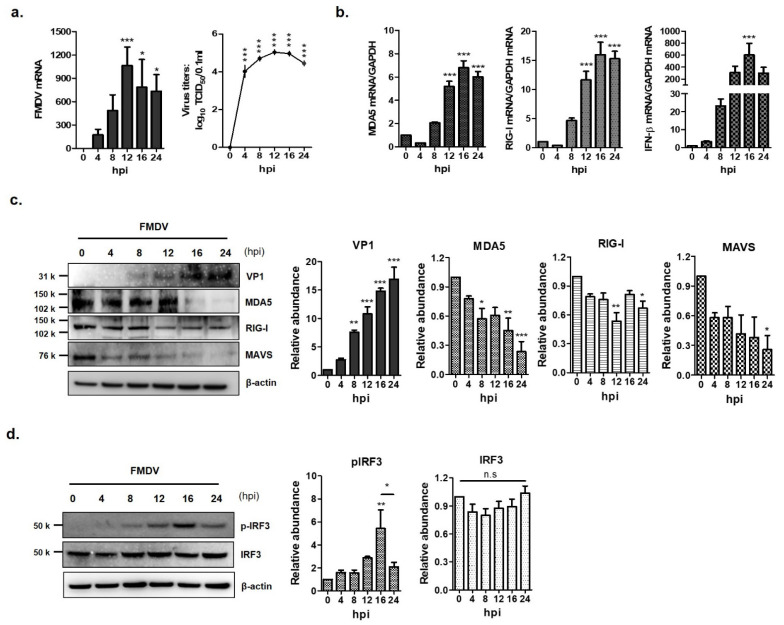
Down-regulation of MDA5 protein expression by foot-and-mouth disease virus (FMDV) infection. The PK-15 cells (1.5 × 10^6^ cells) were infected with FMDV (0.5 MOI) and viral RNA, total RNA and protein were extracted at different times (0, 4, 8, 12 and 24 hpi) after infection. (**a**) The levels of viral RNA were examined by real-time RT-PCR (left panel) and viral titers were determined by TCID50 assay (right panel). (**b**) The levels of MDA5 (left panel), RIG-I and IFN-β (right panel) mRNA in the cell lysates were determined by real-time RT-PCR normalized to GAPDH. (**c**) The levels of endogenous MDA5, RIG-I proteins and VP1 structural protein of FMDV were detected by Western blotting. (**d**) The expression levels of the IRF3 and p-IRF3 proteins were analyzed by Western blotting (left panel). All experiments were performed in triplicate and presented as mean ± SEM (*, *p* < 0.05; **, *p* < 0.01; ***, *p* < 0.001 and n.s., not significant). The abundance of proteins were calculated by Image J software to quantify the band intensity, normalized to β-actin and respectively compared with the 0 hpi.

**Figure 2 cells-10-00271-f002:**
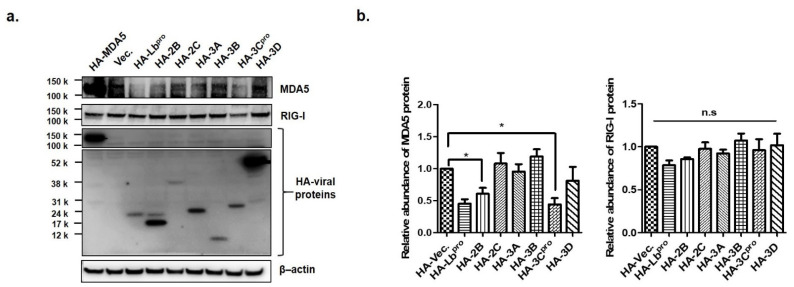
The effects of Lb^pro^ and 3C^pro^ on endogenous MDA5 protein expression. (**a**) HEK293T cells (1 × 10^6^ cells) were transfected with HA (Hemagglutinin)-tagged proteins plasmids (MDA5, Lb, 2B, 2C, 3A, 3B, 3C or 3D) or empty HA-vector plasmid at the amount of 2 μg. Cells were lysed 16 h later and analyzed by Western blotting with anti-HA, MDA5, RIG-I and β-actin antibodies. (**b**) The abundance of MDA5 and RIG-1 proteins were calculated by Image J software to quantify the band intensity, normalized to β-actin and respectively compared with the group of HA-vector. The result is shown of one of the triplicate experiments, presented as mean ± SEM (*, *p* < 0.05 and n.s., not significant).

**Figure 3 cells-10-00271-f003:**
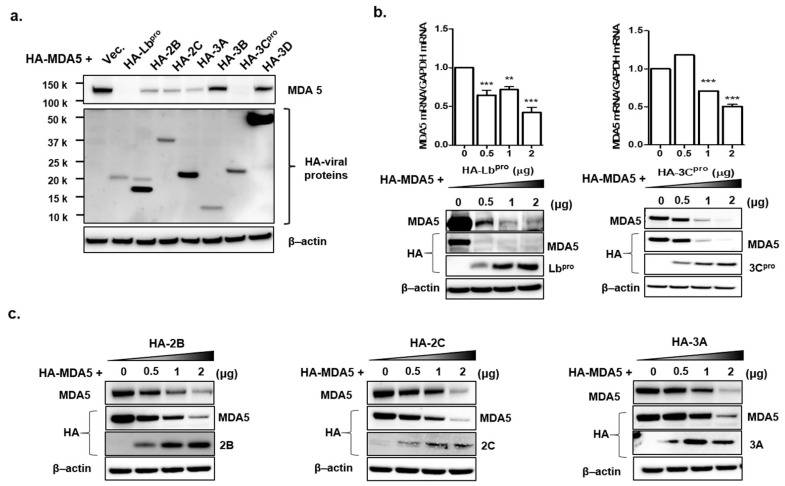
The effects of Lb^pro^ and 3C^pro^ on exogenous MDA5 protein expression. (**a**) HEK293T cells (1 × 10^6^ cells) were co-transfected with HA-tagged MDA5-expressing plasmid (2 μg) and various plasmids expressing HA-tagged viral nonstructural proteins (Lb, 2B, 2C, 3A, 3B, 3C or 3D) or empty HA-vector plasmid (2 μg). The expression of HA-tagged NSP and MDA5 was determined by Western blotting at 16 h. (**b**) HEK293T cells (1 × 10^6^ cells) were transfected with HA-tagged MDA5-expressing plasmid (2 μg) along with different amounts of HA-Lb or 3C expressing plasmid (0, 0.5, 1 or 2 μg). The expression of MDA5 mRNA was detected by real-time RT-PCR normalized to GAPDH (upper panel) and levels of proteins were analyzed by Western blotting with anti-HA, MDA5 and β-actin antibodies at 16 h. The values of mRNA levels are presented as mean ± SEM (**, *p* < 0.01; ***, *p <* 0.001). (**c**) HEK293T cells (1 × 10^6^ cells) were transfected with HA-tagged MDA5-expressing plasmid (2 μg) along with different dose of HA-NSP (2B, 2C or 3A) expressing plasmid (0, 0.5, 1 or 2 μg) for 16 h. The expression of HA-tagged NSP and MDA5 proteins in cell lysates were analyzed by Western blotting with anti-HA, MDA5 antibodies. All the tests were conducted three times independently and data are shown from one of the triplicate experiments.

**Figure 4 cells-10-00271-f004:**
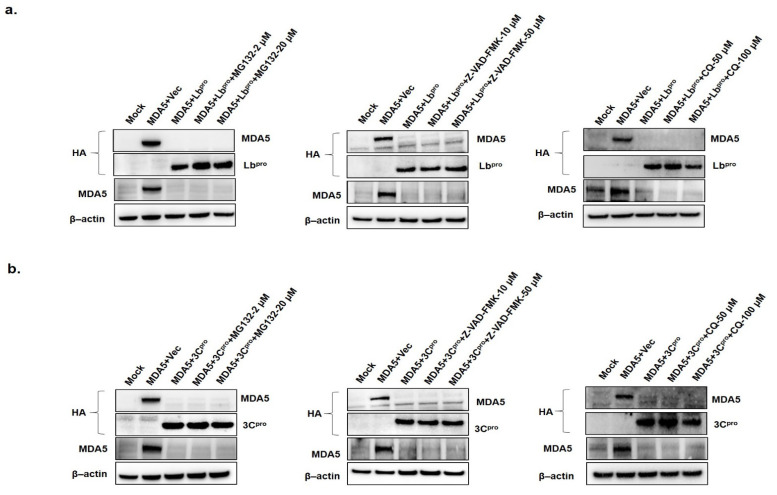
The effects of proteasome, lysosome and caspase inhibitors on Lb^pro^- or 3C^pro^-induced MDA5 reduction. (**a**) HEK293T cells (1.5 × 10^6^ cells) were mock-transfected or were co-transfected with HA-MDA5-expressing plasmid (2 μg) and empty vector (2 μg) or HA-Lb^pro^ expressing plasmids (2 μg). Cells were maintained for 12 h in the presence or absence of the proteasome inhibitor MG132 (2 or 20 μM), the caspase inhibitor Z-VAD-FMK (10 or 50 μM) or lysosome inhibitor chloroquine (CQ; 50 or 100 μM). (**b**) HEK293T cells (1.5 × 10^6^ cells) were mock or co-transfected with HA-MDA5-expressing plasmid (2 μg) and empty vector (2 μg) or HA-3C^pro^ expressing plasmid (2 μg) and maintained in the presence or absence of MG132 (2 or 20 μM), Z-VAD-FMK (10 or 50 μM) or CQ (50 or 100 μM). The protein levels of the above-described experiments were determined by Western blotting with anti-HA, MDA5 and β-actin antibodies. All the data were repeated in three independent experiments and data from one of the triplicate experiments are shown.

**Figure 5 cells-10-00271-f005:**
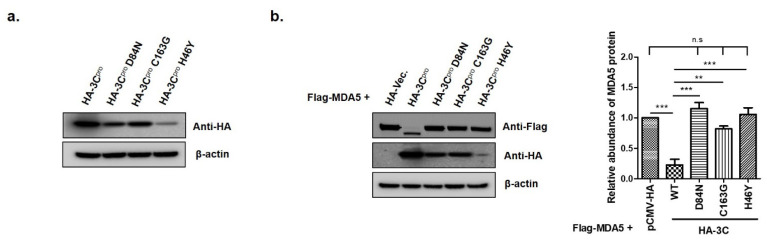
The effects of catalytic residues H46, D84 and C163 of the 3C^pro^ active site on FMDV 3C^pro^-induced MDA5 protein reduction. (**a**) HEK293T cells (1.5 × 10^6^ cells) were transfected with HA-tagged empty vector, 3C^pro^ or individual mutants of 3C^pro^ (2 μg) for 12 h. (**b**) HEK293T cells (1.5 × 10^6^ cells) were co-transfected with HA-tagged, MDA5-expressing plasmid (2 μg) and various plasmids expressing 3C^pro^ or individual mutants of 3C^pro^ (2 μg) for 12 h. The abundance of MDA5 proteins was calculated by Image J software to quantify the band intensity, normalized to β-actin and respectively compared with the group of HA-Vec. or HA-3C^pro^. The result from one of the triplicate experiments are shown and presented as mean ± SEM (**, *p* < 0.01; ***, *p* < 0.001 and n.s., not significant).

**Figure 6 cells-10-00271-f006:**
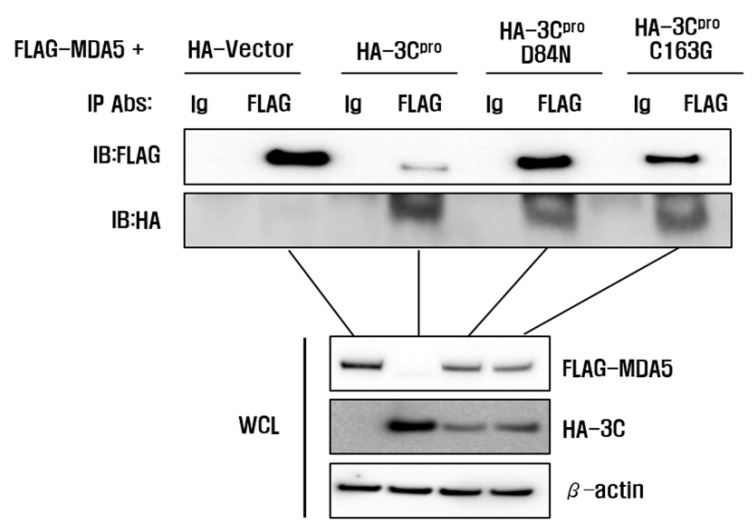
Interaction between 3C^pro^ and MDA5. HEK293T cells grown in 10-cm dishes were co-transfected with FLAG-MDA5 plasmid (8 μL) and the indicated plasmids (8 μL). The cells were lysed at 10 h and immunoprecipitated with mouse anti-FLAG antibody or mouse normal IgG antibody and then subjected to western blotting with anti-FLAG or anti-HA antibodies. Whole cell lysates (WCL) were analyzed by immunoblotting (IB) using anti-FLAG or anti-HA antibodies. All the data were repeated in three independent experiments and data from one of the triplicate experiments are shown.

**Table 1 cells-10-00271-t001:** The real-time RT-PCR primers for target gene expression.

Type	Primer	Sequence (5′->3′)
Porcine	MDA5-F	GTAGGAGTCAAAGCCCACCA
	MDA5-R	GACTTCTCTTTGTTCATTCTGTGTC
	RIG-I-F	CTGCAGACATGGGATGAAGCA
	RIG-I-R	TTATCAGGCACAGGTTCTGGTTT
	IFN-βF	GGCTGGAATGAAACCGTCAT
	IFN-β-R	TCCAGGATTGTCTCCAGGTCA
	GAPDH-F	ACATGGCCTCCAAGGAGTAAGA
	GAPDH-R	GATCGAGTTGGGGCTGTGACT
Human	MDA5-F	GCTGAAGTAGGAGTCAAAGCCC
	MDA5-R	CCACTGTGGTAGCGATAAGCAG
	RIG-I-F	CACCTCAGTTGCTGATGAAGGC
	RIG-I-R	GTCAGAAGGAAGCACTTGCTACC
	IFN-β-F	TTGTTGAGAACCTCCTGGCT
	IFN-β-R	TGACTATGGTCCAGGCACAG
	GAPDH-F	GAGTCAACGGATTTGGTCGT
	GAPDH-R	GACAAGCTTCCCGTTCTCAG

## Data Availability

Not applicable.

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
