# Peer review of "Foot-and-Mouth Disease Virus Evades Innate Immune Response by 3C-Targeting of MDA5"

_cells, 2021, doi:10.3390/cells10020271_

Round 1
Reviewer 1 Report
January 5th, 2021
Review: cells-1065379
Foot-and-mouth disease virus evades innate immune response by 3C-targeting of MDA5
In this manuscript Kim et. al. described the interplay between MDA5 and FMDV, focusing on 3Cpro. The research field is of great interest since understanding mechanisms used by FMDV to block immune response could be applied to develop countermeasure strategies to control this devastating large animal disease. The data and experimental design are scientifically correct, although there are some things that need to be clarify. The authors described a reduction in expression MDA5 while the mRNA increases but they do not see a degradation product. The degradation product is actually no detected in any of the experiments described, even when the protein was exogenously expressed by plasmid transfection. The authors need to thoroughly discuss this issue and try to explain why the protein is not expressed or disappearing when there is no degradation product. Another important point missing in the manuscript is the lack of experiments in which the authors show direct interaction between 3Cpro and MDA5. If the authors think that the MDA5 reduction is due to indirect interactions, the authors should try to show what other proteins get degraded. For instance, Rodriguez-Pulido demonstrated degradation of LGP2 by Lpro as way to interact with immune response.
Other points:
- WB images showed that the efficiency of transfection is different for the different FMDV proteins. Could this be responsible for the difference in the effect on MDA5?
- Figure 1 shows wb detection of MAVS. However, this was not included in the M&M section neither is given any importance in the discussion. Could you explain why?
- Figure 2b: In the HA-3C wb in the double band detected a product of degradation? Why is this the only time is detected?
- Figure 4a: Why in the wb of CQ treated cells there is a band for MDA detection on the Mock transfected cells?
- The authors discussed that they do not include data on Lpro catalytic effect con MAD5 because it has bee previously reported. Could they elaborate on why there is a function redundancy for FMDV with two proteins showing the same effect?
Reviewer 2 Report
The authors examined the status of MDA5 in cultured kidney cells after foot-mouth disease virus (FMDV) infection. They found that FMDV infection decreases MDA5 protein expression due to 3C proteinase activity.
This MS is very interesting and informative article. I enjoy it. However, several concerns are arisen before reconsideration of its suitability of publication.
Major,
1. They used three different cell types in their experiment. For example, the results described in the figure 1 were obtained from PK-15 cells whereas the results described in the figures 2, 3, 4 and 5 were obtained from HEK293T cells. It is needed to explain clearly why the authors used different cell types in each experiment.
2. After FMDV infection, the changes of MDA5 expression between mRNA and protein level were apparently different. This issue should be explained clearly.
3. It is needed to perform additional experiment using siRNA method against 3C proteinase. Then, the authors should prove their novel findings in this MS.
4. How about the implication of TLR3 in FMDV infection? Please clarify this issue.
Round 2
Reviewer 1 Report
The authors have addressed all the comments from the previous review. The article is now ready for publication.
Reviewer 2 Report
I understand the authors' response.
However, it is nice to add brief sentences regarding the reason why the authors used different cell types in each experiment. Also, some brief sentences regarding the answer to my concerns should be added in their revised MS.
